

# Nucleolin myocardial-specific knockout exacerbates glucose metabolism disorder in endotoxemia-induced myocardial injury

Yuting Tang[1,2,3,*], Leijing Yin[1,2,3,*], Ludong Yuan[1,2,3], Xiaofang Lin[1,2,3] and Bimei Jiang[1,2,3]

[1] Department of Pathophysiology, Xiangya School of Medicine, Central South University, Changsha, Hunan, China
[2] Sepsis Translational Medicine Key Lab of Hunan Province, Central South University, Changsha, Hunan, China
[3] National Medicine Functional Experimental Teaching Center, Central South University, Changsha, Hunan, China
* These authors contributed equally to this work.

Corresponding author
Bimei Jiang, jiangbimei@csu.edu.cn

## ABSTRACT

**Background:** Sepsis-induced myocardial injury, as one of the important complications of sepsis, can significantly increase the mortality of septic patients. Our previous study found that nucleolin affected mitochondrial function in energy synthesis and had a protective effect on septic cardiomyopathy in mice. During sepsis, glucose metabolism disorders aggravated myocardial injury and had a negative effect on septic patients.

**Objectives:** We investigated whether nucleolin could regulate glucose metabolism during endotoxemia-induced myocardial injury.

**Methods:** The study tested whether the nucleolin cardiac-specific knockout in the mice could affect glucose metabolism through untargeted metabolomics, and the results of metabolomics were verified experimentally in H9C2 cells. The ATP content, lactate production, and oxygen consumption rate (OCR) were evaluated.

**Results:** The metabolomics results suggested that glycolytic products were increased in endotoxemia-induced myocardial injury, and that nucleolin myocardial-specific knockout altered oxidative phosphorylation-related pathways. The experiment data showed that TNF-α combined with LPS stimulation could increase the lactate content and the OCR values by about 25%, and decrease the ATP content by about 25%. However, interference with nucleolin expression could further decrease ATP content and OCR values by about 10–20% and partially increase the lactate level in the presence of TNF-α and LPS. However, nucleolin overexpression had the opposite protective effect, which partially reversed the decrease in ATP content and the increase in lactate level.

**Conclusion:** Down-regulation of nucleolin can exacerbate glucose metabolism disorders in endotoxemia-induced myocardial injury. Improving glucose metabolism by regulating nucleolin was expected to provide new therapeutic ideas for patients with septic cardiomyopathy.

## INTRODUCTION

Sepsis, a life-threatening organ dysfunction caused by a dysregulated host response to infection, is one of the most common causes of death in critically ill patients (*Martin et al., 2019*; *Singer et al., 2016*). Cardiovascular dysfunction in patients with sepsis leads to a further increase in mortality, which is also called septic cardiomyopathy or sepsis-induced myocardial injury (*Fernandes & de Assuncao, 2012*). Septic patients with confirmed or suspected infection were usually associated with endotoxemia (*Dickson & Lehmann, 2019*). Therefore, endotoxemia was often used to mimic the acute inflammatory response associated with early sepsis. Myocardial tissue needs a constant ATP supply and stored ATP can't sustain for long, making regulation of energy metabolism essential.
The metabolism of free fatty acids (FFA), glucose, ketone bodies (KB), and amino acids (AA) are the source of ATP production in myocardial tissue (*Wende et al., 2017*). Oxidative phosphorylation (OXPSH) in mitochondria provides 95% of ATP to the heart in normal physiological conditions.

Mitochondrial dysfunction and bioenergetic failure have long been recognized as important pathophysiologic mechanisms of multi-organ dysfunction in septic patients. The uptake of energy substrates (FFA, glucose, KB) in the heart was frequently reduced in sepsis (*Dhainaut et al., 1987*). Studies have shown that sepsis enhanced the glycolytic metabolism, which correlated with high mortality from sepsis-induced myocardial injury; however, inhibition of glycolytic metabolism using 2-deoxy-D-glucose (2-DG) improved cardiac function and survival in septic mice (*Zheng et al., 2017*). Myocardial injury in sepsis can be attenuated by targeting TLR-mediated inflammatory responses and then the prognosis of sepsis improved (*Gao et al., 2012*). It is suggested metabolic reprogramming played a key role in host defense and inflammation. Both TLRs and inflammatory cytokines were involved in the regulation of metabolic reprogramming from oxidative phosphorylation to aerobic glycolysis (*O'Neill & Pearce, 2016*; *Vander Heiden, Cantley & Thompson, 2009*). Furthermore, intermediates produced by aerobic glycolysis and metabolism can in turn regulate immune function (*Loftus & Finlay, 2016*). Studies have shown that sepsis-induced cardiac dysfunction can be attributed to an enhanced glycolytic metabolism, however, early modulation of these alterations could be an appropriate approach to treating sepsis (*Liu et al., 2022*). Therefore, the role played by altered glucose metabolism in sepsis-induced myocardial injury deserves to be investigated.

Nucleolin is a multi-functional protein found mainly in the nucleolus and can be involved in a variety of pathophysiologic processes (*Jia et al., 2017*). In previous studies, nucleolin has been shown to play a protective role in ischemic cardiomyopathy, adriamycin-induced myocardial injury, and sepsis-induced myocardial injury (*Chen et al., 2020*; *Jiang et al., 2013*; *Jiang et al., 2019*; *Sun et al., 2018*). However, little was known about the exact mechanism by which nucleolin protected the myocardium from sepsis-induced myocardial injury. In the previous research, we explored that down-regulation of nucleolin could lead to decreased ATP production in cardiomyocytes during LPS-induced myocardial injury (*Yin et al., 2023a*), and thus we hypothesized that nucleolin could affect cardiomyocyte energy metabolism and probably mainly on glucose metabolism during

sepsis. Therefore, we explored the effect of nucleolin on cardiac glucose metabolism during endotoxemia-induced myocardial injury.

Metabolomics technology, which qualitatively and quantitatively analyzes energy metabolites and searches for the relative relationship between metabolites and physiopathological alterations, has been widely used in the fields of new drug discovery, disease diagnosis, and personalized therapy (*Cao et al., 2023*; *van der Hooft et al., 2020*). Thus, we collected cardiac tissues from an endotoxemic mouse model and examined the effect of nucleolin cardiac-specific knockout on glucose metabolism through untargeted metabolomics. We then verified the metabolites by experiments in a TNF-α combined with LPS-stimulated H9C2 cells. Final conclusions were drawn to validate the effect of nucleolin on cardiac glucose metabolism in endotoxemia-induced myocardial injury.

## MATERIALS AND METHODS

### Animals and treatment

$NCL^{flox/flox}$ mice and Myh6-Cre transgenic mice were purchased from Guangzhou Cyagen Biotechnology Co. Nucleolin cardiac-specific knockout mice ($NCL^{flox/flox; \ Myh6-Cre}$) were constructed as previously described (*Yin et al., 2023a*). The mice were housed in an SPF-grade barrier system with light/dark cycle switching every 12 h and free access to water and food throughout the experimental period. A mouse model of endotoxemia was constructed by intraperitoneal injection of LPS (10 mg/kg) as a way to mimic the acute inflammatory response in the early stage of sepsis (*Dickson & Lehmann, 2019*; *Yin et al., 2023b*). Then, the male mice about 8–12 weeks old were randomly divided into two groups, the control group, and the LPS-injected 24-h group. The LPS-injected group was injected intraperitoneally using LPS (10 mg/kg), and the control group was injected with an equal amount of saline, followed by 1 ml of saline injected subcutaneously for rehydration. Animals were then observed at 12-h intervals, and cares were taken to keep the experimental animals safe and warm. During the whole experiment, only necessary drug injection experiments were carried out on the experimental animals, and disinfection and pacification were carried out before injection to ensure that the experimental animals were free from pain and fear. Based on statistical requirements, we selected as few animals as possible for grouping. At the end of the experiment, in keeping with our commitment to animal ethics, we evacuated the mice by cervical dislocation after 2% isoflurane anesthesia by inhalation due to the need to obtain myocardial tissue for experimental studies. Animal handling procedures were performed in strict accordance with the animal welfare guidelines established by the World Organization for Animal Health and the Chinese National Guidelines for Animal Testing. The study protocol was approved by the Department of Animal Experimentation, Medical Ethics Committee of the Third Xiangya Hospital, Central South University (2019-S218).

### Reagents

The LPS (L2880) was purchased from Sigma-Aldrich (St. Louis, MO, USA).
The TNF-alpha protein (rat, recombinant) was purchased from Sino Biological (80045-RNAE, Beijing, China).

## Cell culture

H9C2(2-1) cell lines were obtained from American Type Culture Collection (CRL-1446, ATCC, Manassas, VA, USA), and cultured in DMEM contained with 10% fetal bovine serum, 10% streptomycin (100 μg/mL) and 10% penicillin (100 U/mL). Overexpression or interference with the expression of nucleolin in the H9C2 cell line was performed as described previously (Yin et al., 2023a).

## The ATP content and the Lactate quantification assay

The adenosine triphosphate (ATP) content was detected according to the manufacturer's instructions by the ATP Assay kit (Beyotime, Shanghai, China) as described previously (Yin et al., 2023b). Lactate accumulation in cultures was assessed using a lactate estimation kit (Nanjing Jiancheng, China) according to the manufacturer's instructions as described previously (Tan et al., 2021). The concentrations of cellular proteins used for detection were measured and used for normalization.

## Oxygen consumption rate

H9C2 cells were spread on XFe cell culture plates with 7,000 cells per well in the middle of six wells, and one well at each end was left as a blank control. The cell culture plates were incubated overnight at 37 °C in a 5% $CO_2$ concentration environment. The probe plate was hydrated by adding sterile water and incubated overnight in a $CO_2$-free 37 °C incubator, and the calibration solution was also incubated overnight in a $CO_2$-free 37 °C incubator. After removing the cell culture plate from the medium and washing it once with the detection solution, the liquid in the cell culture plate was replaced with the calibration solution that was incubated overnight in a $CO_2$-free 37 °C incubator for a minimum of 45 min to 1 h. The assay was then performed on a Seahorse machine. During the experiment, the machine automatically added oligomycin, FCCP, and rotenone/antimycin sequentially to the cell culture plates at different times to obtain the real-time oxygen consumption rate. The final concentrations of oligomycin, rotenone/antimycin, and FCCP were 1.5, 0.5, and 0–2 μM, respectively. The relevant calculation method and the timing of drug addition were controlled by a pre-set computer program, which was also described in the Supplemental Material section. After the completion of the experiment, cell culture wells were calibrated for quantification by adding 20 μl of RIPA lysate containing PMSF per well to detect the protein concentration, and the data were analyzed by Agilent ATP Assay Report Generator (Agilent, Santa Clara, CA, USA) and then were normalized to the cellular protein level.

The seahorse data calculations were described as follows: basal respiration was determined as the value of last rate measurement before first injection minus the value of non-mitochondrial respiration rate; maximal respiration was determined as the value of maximum rate measurement after FCCP injection minus the value of non-mitochondrial respiration; ATP production was determined as the value of last rate measurement before oligomycin injection minus the value of minimum rate measurement after oligomycin injection; spare respiratory capacity was determined as the value of maximal respiration minus the value of basal respiration.

## Sample processing

A total of 80% methanol solution (500 μl) was added to the liquid nitrogen ground myocardial tissue samples (100 mg). After vortexing and mixing, the mixture was placed in an ice bath for 5 min and centrifuged at $15,000 \times g$ for 20 min at 4 °C. Then the supernatant was diluted with mass spectrometric water to 53% methanol. Finally, the supernatant was centrifuged at $15,000 \times g$ for 20 min at 4 °C and subjected to LC-MS analysis.

## Metabolite identification

The downstream data (. raw) file was imported into the CD database search software. After data processing, peak extraction, peak area quantification, and reintegration of target ions, the prediction was performed. The data were compared with mzCloud (https://www.mzcloud.org/), mzVault, and Masslist databases, and the background ions were removed and normalized. Finally, the data were obtained and analyzed.

## Statistical analysis

The data were expressed as mean ± standard error, and the Shapiro-Wilk test was used to detect the normal distribution of the data. Comparisons between multiple groups were performed using one-way ANOVA or two-way ANOVA, and further comparisons between two groups were performed using the SNK-q test. Student's t-test was used to compare the two groups of data. Predictive analysis for metabolite identification was performed using a multiparameter joint predictive model, first by logistic regression analysis and then by ROC curve analysis. Metabolomics data were analyzed using the Omsicshare website. $p < 0.05$ was considered statistically significant. All data were statistically analyzed using Graphpad Prism, MetaboAnalyst.

# RESULTS

## Myocardial-specific nucleolin knockout altered myocardial tissue metabolomics in mice during endotoxemia

In this study, untargeted metabolomics was performed in the NCL[flox/flox] group (WT group, $n = 4$), the NCL[flox/flox] + LPS group (WT_LPS group, $n = 4$) and the NCL[flox/flox; Myh6-Cre] + LPS group (KO_LPS group, $n = 5$), to investigate the effects of nucleolin cardiac-specific knockout on glucose metabolism during endotoxemia-induced myocardial injury. Metabolomics was based on mass spectrometry (MS), which utilized a combination of positive ion mode (POS) and negative ion mode (NEG) to achieve better detection results.

The reliability of the data was verified through the principal component analysis (PCA) and quantity control (QC). As shown in Figs. 1A and 1B, the QC samples (red crosses in the figure) were closely distributed in the PCA plots of POS and NEG, which confirmed that the error of this experiment was small and the data were reliable. To further analyze this metabolomics, PCA analysis was performed on the metabolomics samples among these groups (Figs. 1C–1F). These results showed that there was significant variation in the metabolomics data between the two groups, while the within-group variability was

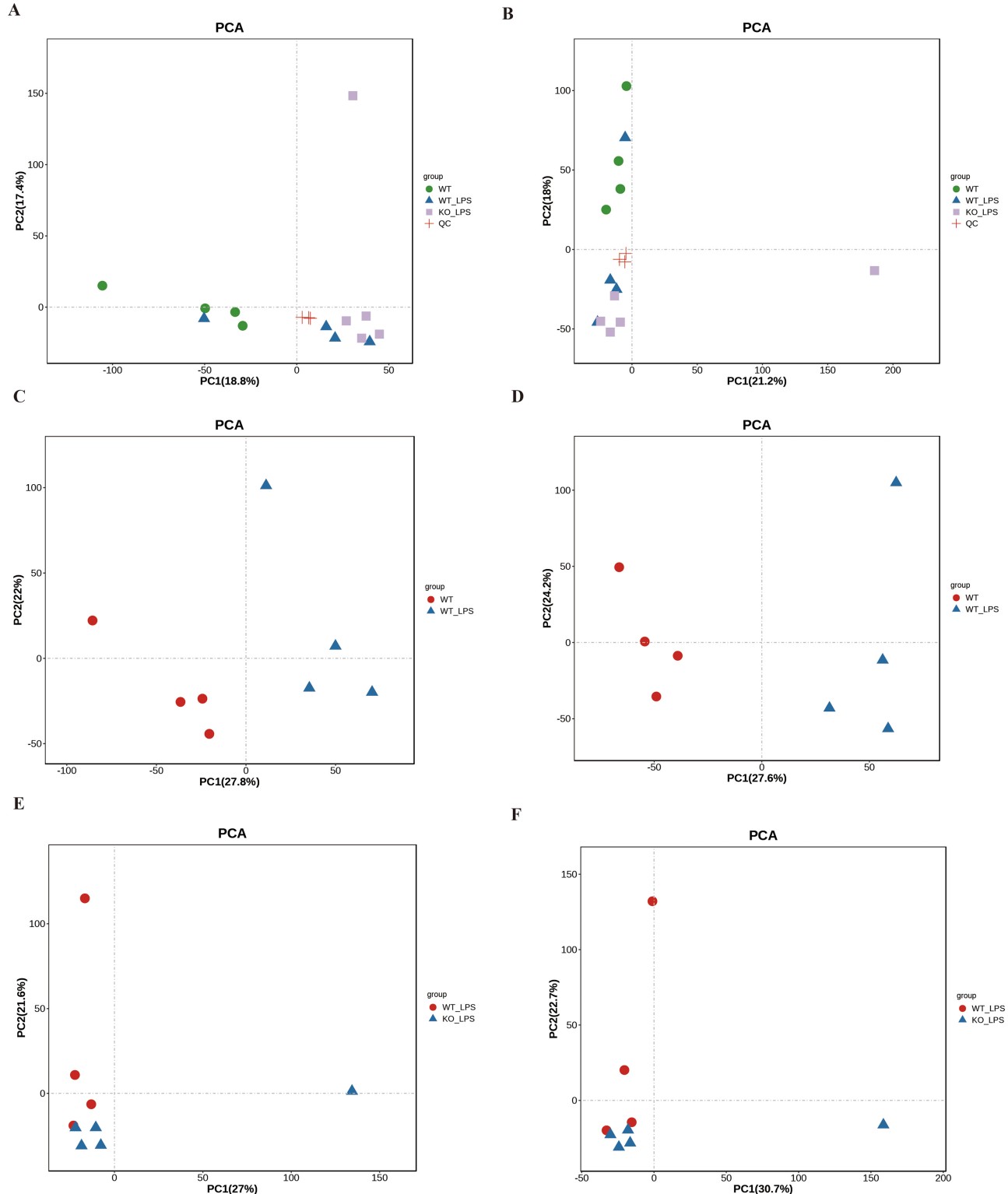

**Figure 1 Quality control and PCA analysis of metabolomics samples.** (A and B) POS (Figure A) and NEG (Figure B) data were analyzed by PCA for their quality control results. (C and D) PCA analysis of myocardial metabolomics POS (Figure E) and NEG (Figure F) data in the WT group versus the WT _LPS group. (E and F) PCA analysis of myocardial metabolomics POS (Figure E) and NEG (Figure F) data in WT_LPS group versus KO_ LPS group. The tight distribution of samples in each of the two groups suggested intergroup variability in metabolites and low intragroup variability.

relatively low. To better understand the differences and identify specific metabolites between the groups, the partial least squares discrimination analysis (PLS-DA) was used, and the results were consistent with those in the PCA analysis (Figs. 2A to 2D). Next, volcano plots were used to analyze the metabolomics data, as shown in Figs. 3A to 3D, demonstrating the two differential metabolite profiles and their VIP (variable impact importance factor) values. After analyzing the volcano plots, we counted the number of metabolite differences in each group. As shown in Figs. 3E and 3F, the myocardial-specific knockout of nucleolin and LPS-induced myocardial injury can cause significant changes in the number of myocardial metabolites.

The study aimed to gain a further understanding of the differential metabolites by statistically calculating the abundance of each metabolite in each group. Samples that obtained a VIP value greater than 1 were analyzed and tabulated, as they were considered to be significantly different. When comparing the WT group to the WT_LPS group using the NEG mode assay, it was observed that there was a decrease in pyruvate in response to LPS stimulation. Similarly, when comparing the WT_LPS group to the KO_LPS group using the NEG mode assay, it was observed that there was a decrease in malate as a product of the tricarboxylic acid cycle, which suggested that the tricarboxylic acid cycle might be inhibited.

## Metabolic pathway analysis of myocardial differential metabolites

We have identified the metabolic differences between the WT group, WT_LPS group, and KO_LPS group. The differentially expressed metabolites were then annotated using the KEGG database to determine the biological processes in which these metabolites are primarily involved. The results, as shown in Fig. 4A, suggested that the metabolic alterations among these three groups were mainly involved in lipid metabolism, carbohydrate metabolism, and amino acid metabolism pathways. Among the KEGG annotations of human disease types, bacterial and parasitic infection pathways, and cardiovascular diseases were more enriched, which also suggested that the endotoxemic model could better mimic the pathogen infection-induced injury model.

To more intuitively show the specifics of the relevant enriched pathways, KEGG enrichment bubble plots for differential metabolites were used to perform statistical analysis. It's suggested that fatty acid biosynthesis, unsaturated fatty acid biosynthesis, and pyruvate metabolism pathways might be altered in endotoxemia (Fig. 4B). It is also suggested in Fig. 4C that the myocardial-specific knockout of nucleolin could significantly affect the metabolic processes of purine metabolism, biotin metabolism, fatty acid biosynthesis, and alanine biosynthesis. To better understand the metabolic changes, the metabolite concentration changes by Metabolite Set Enrichment Analysis (MSEA) were also analyzed. The results suggested that the metabolite concentration was higher in the tricarboxylic acid cycle, ketone body metabolism, Warburg effect metabolic processes, and the fatty acid β-oxidation between the WT group and WT_LPS group (Fig. 5A and 5B), the WT_LPS group and the KO_LPS group (Fig. 5C and 5D).

Based on the results of the differential metabolite abundance analyses, KEGG analysis, and MSEA metabolite analyses, it can be assumed that during endotoxemia, there was an

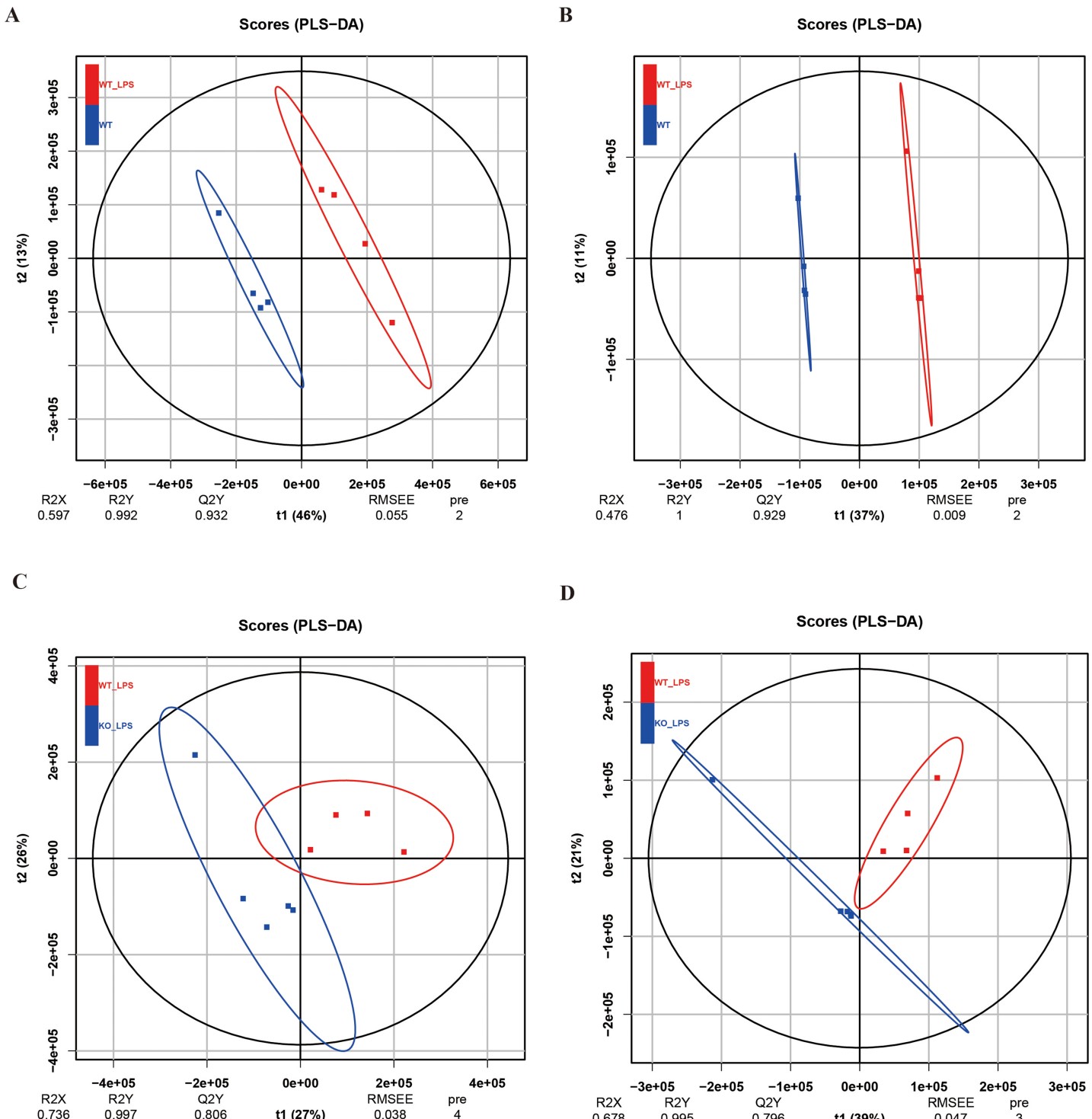

**Figure 2  PLS-DA analysis of metabolomics samples.** (A and B) PLS-DA analysis of myocardial metabolomics POS (Figure A) and NEG (Figure B) data in the WT group versus the WT_LPS group. The Q2 values obtained for both PLS-DA of POS and NEG were greater than 0.9, indicating a significant difference between the two groups of metabolites. (C and D) PLS-DA analysis of myocardial metabolomics POS (Figure C) and NEG (Figure D) data in WT_ LPS group versus KO_LPS group. The Q2 values of the results in the POS and NEG groups were 0.806 and 0.796, respectively, showing the variability between the metabolomics data of the two groups.

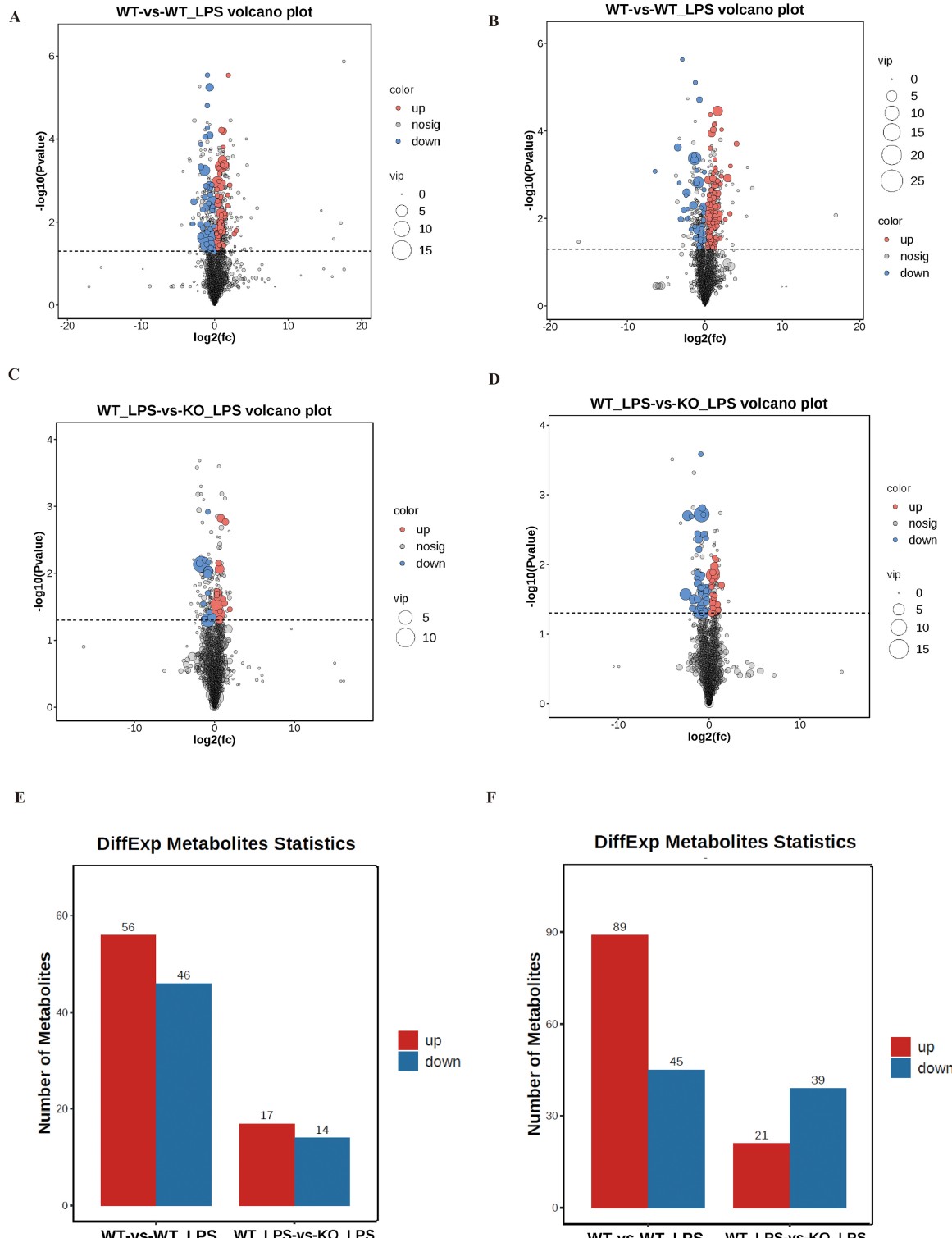

**Figure 3 Identification of specific differential metabolites between metabolomics samples.** (A and B) Volcano plot of metabolite differences between WT group and WT_LPS group in POS and NEG mode. The up-regulated and down-regulated metabolites and the corresponding VIP values were shown, respectively. (C and D) Volcano plot of metabolite differences between WT_LPS group and KO_LPS group in POS and NEG mode. (E and F) The number of metabolite differences between groups in POS mode and NEG mode.

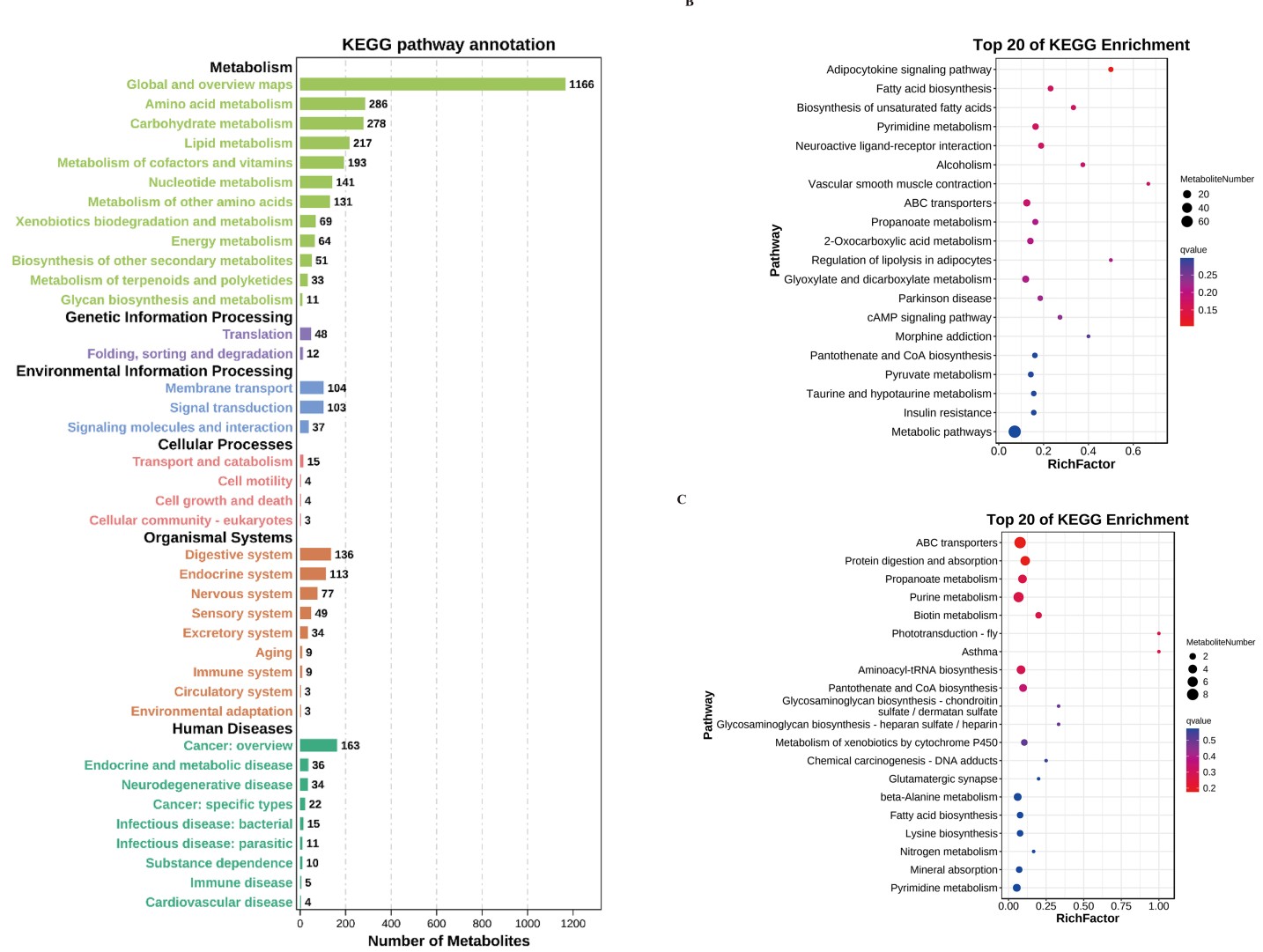

**Figure 4 Metabolic pathway analysis of myocardial differential metabolites.** (A) KEGG annotation of myocardial differential metabolites. (B) KEGG enrichment bubble plots of differential metabolite between the WT group and WT_LPS group. The 20 metabolic pathways with the highest difference values were plotted as vertical coordinates and the degree of enrichment as horizontal coordinates. (C) KEGG enrichment bubble plots of differential metabolite between WT_LPS group and KO_LPS group.

increase in fatty acid synthesis and a decrease in pyruvate levels in myocardial tissues. This could be due to significant changes in the tricarboxylic acid cycle or an impact on glycolytic function. The reduction of malate levels after the nucleolin myocardial-specific knockout indicated that the tricarboxylic acid cycle and fatty acid biosynthesis were significantly affected. Moreover, the higher abundance values of Warburg effect-related metabolites suggest that glycolytic function might be highly affected.

## The alteration of glucose metabolism during TNF-α combined with LPS stimulation in H9C2 cells

Metabolomics showed that LPS stimulation and nucleolin knockout might lead to enhanced glycolysis in myocardial tissue. To verify these results, we continued the study *in*

**A**

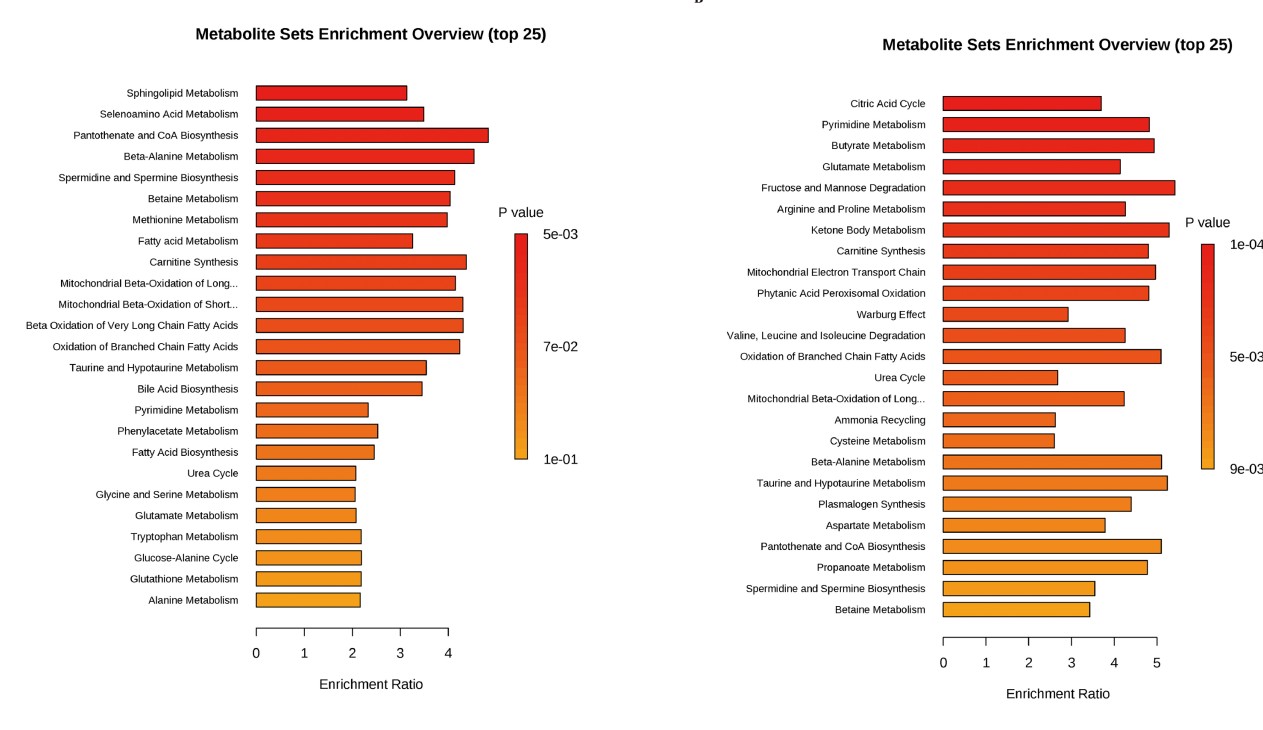

**B**

**C**

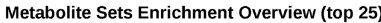

**D**

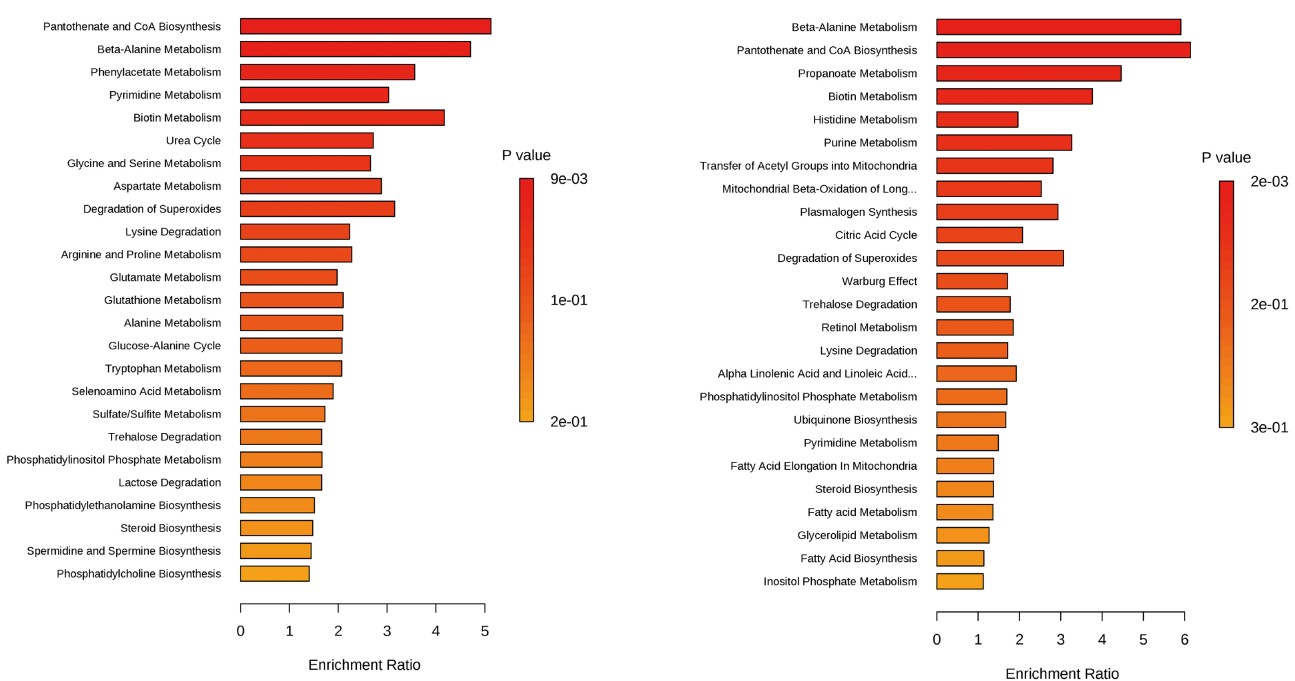

**Figure 5** **Metabolite abundance was analyzed by MSEA analysis.** (A and B) Metabolite abundance in MSEA analysis of differential metabolites between WT group and WT_LPS group in POS mode and NEG mode. (C and D) Metabolite abundance in MSEA analysis of differential metabolites between WT_LPS group and KO_LPS group in POS mode and NEG mode. The vertical coordinate of the MSEA analysis result was the name of the metabolic set, and the horizontal coordinate indicated the degree of enrichment, and the top 25 of the degree of enrichment were selected for analysis in this study.

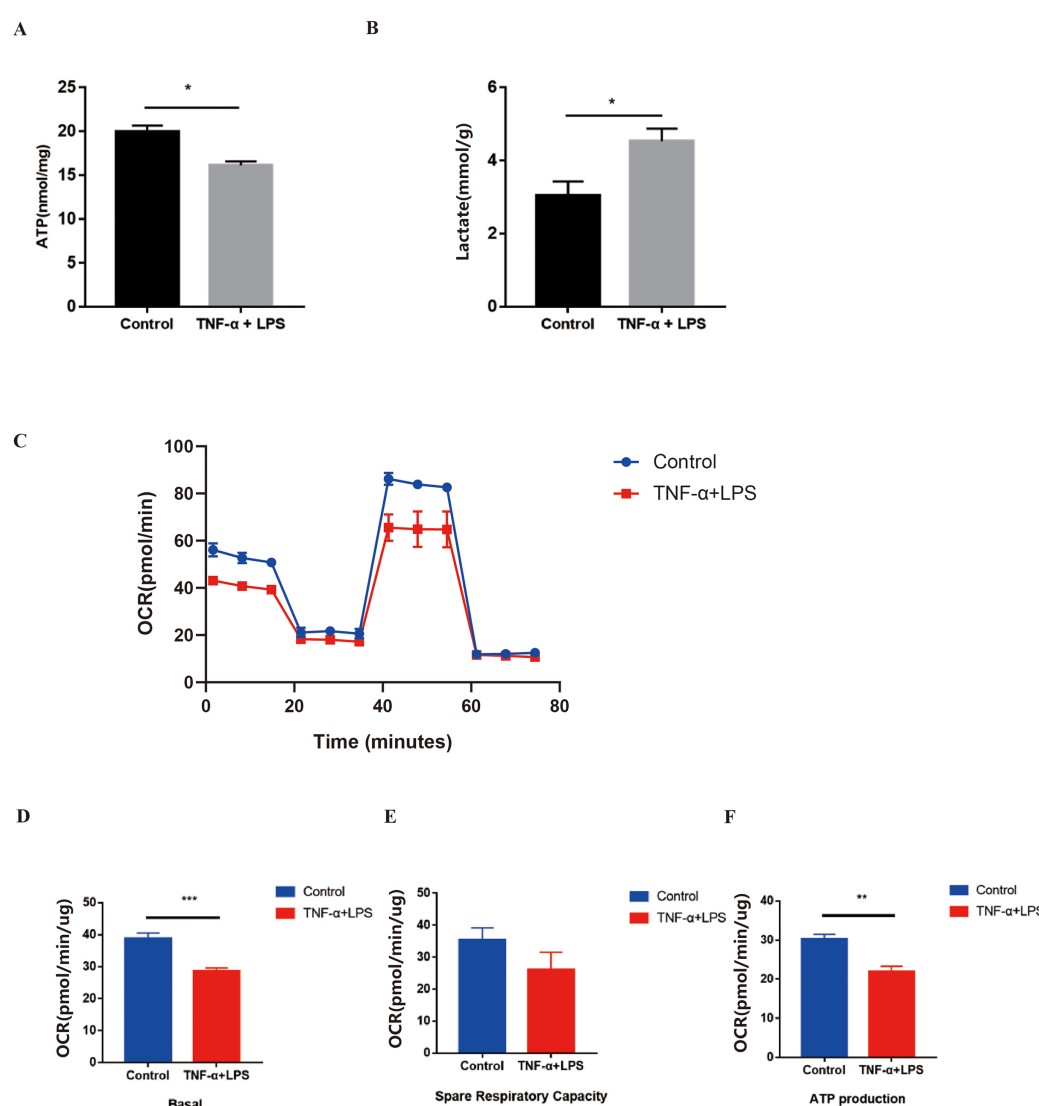

**Figure 6 The alteration of glucose metabolism during TNF-α combined with LPS stimulation in H9C2 cells.** (A) ATP content of different groups were detected by the ATP assay kit. The concentration of ATP was converted to nmol per mg protein. (B) The lactate levels in cell supernatants were detected by the Lactic Acid assay kit. (C) Continuous dynamic changes of OCR after TNF-α combined with LPS stimulation in H9C2 cells. The error bars are sometimes smaller than the symbol. (D–F) The basal OCR value, spare respiratory capacity, and ATP production after TNF-α combined with LPS stimulation in H9C2 cells. $^*p < 0.05$, $^{**}p < 0.01$, $^{***}p < 0.005$; $n \geq 3$ in each group.

*vitro* by using H9C2 cells. The study used TNF-α (50 ng/ml) combined with LPS (1 μg/ml) to stimulate H9C2 cells for 6 h to mimic the inflammatory hyper-responsive state of early endotoxemia. The data showed that after stimulation of H9C2 cells with TNF-α combined with LPS, the cellular supernatant lactate level was increased by about 25% and the ATP content was decreased by about 25% (Figs. 6A–6B), suggesting that energy metabolism was impaired and glycolysis was elevated in H9C2 cells. To better detect the overall oxidative phosphorylation level in H9C2 cells, we further detected the cellular oxygen consumption

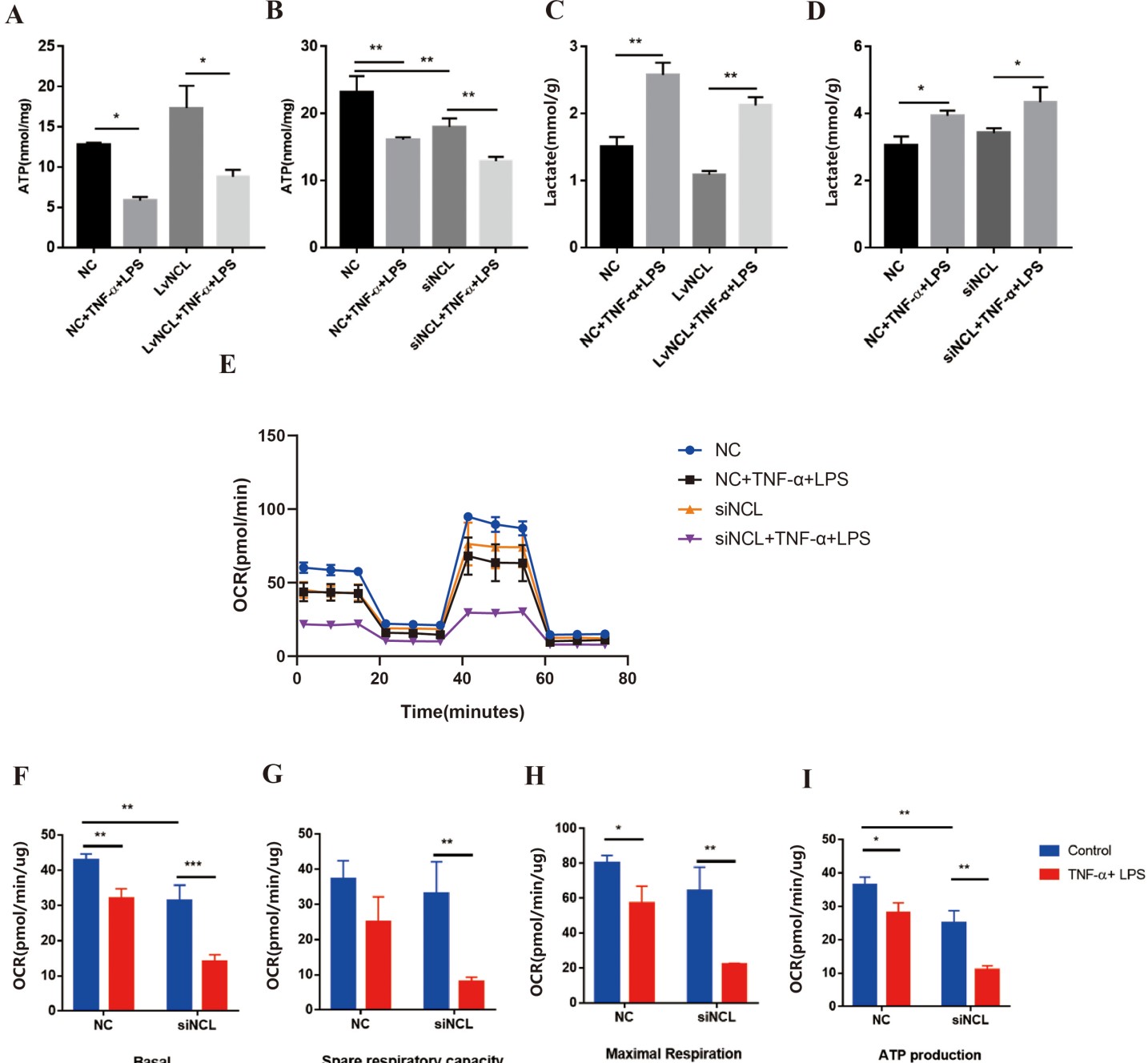

**Figure 7 The effect of nucleolin on glucose metabolism during TNF-α combined with LPS stimulation in H9C2 cells.** (A and B) ATP content of different groups were detected by the ATP assay kit after nucleolin overexpression (LvNCL) or nucleolin interference (siNCL) in the H9C2 cell. The concentration of ATP was converted to nmol per mg protein. (C and D) The lactate levels in cell supernatants were detected by the Lactic Acid assay kit after nucleolin overexpression (LvNCL) or nucleolin interference (siNCL) in the H9C2 cell. (E) Continuous dynamic changes of OCR after TNF-α combined with LPS stimulation in the nucleolin interference (siNCL) groups or control groups. (F–I) The basal OCR value, spare respiratory capacity, and ATP production after TNF-α combined with LPS stimulation of the nucleolin interference (siNCL) groups or control groups. $^{*}p < 0.05$, $^{**}p < 0.01$, $^{***}p < 0.005$; $n \geq 3$ in each group.

rate (OCR) in H9C2 cells through the Seahorse assay. The results showed that the real-time continuous OCR value of H9C2 cells was reduced after TNF-α combined with LPS stimulation (Fig. 6C). The basal OCR value and ATP production capacity were reduced by about 25% and the reserve respiratory capacity was partially reduced. (Figs. 6D–6F). These data suggested that oxidative phosphorylation was impaired in H9C2 cells after TNF-α combined with LPS stimulation.

### The effect of nucleolin on glucose metabolism during TNF-α combined with LPS stimulation in H9C2 cells

The metabolomics analysis has suggested that nucleolin cardiac-specific knockout could affect the metabolic level of myocardial tissue, and we now proceeded to validate it at the cellular level. Compared to the control group, overexpression of nucleolin promoted ATP synthesis and reduced lactate levels, providing a protective effect. In addition, nucleolin overexpression partially reserved the decrease in ATP content and the increase in lactate levels during TNF-α combined with LPS stimulation in H9C2 cells by about 20% (Fig. 7A). In contrast to the partial recovery observed in ATP content after overexpression of nucleolin, small interfering RNA (siRNA) interference led to a decrease in nucleolin expression and ATP content by approximately 10% in H9C2 cells, both with and without TNF-α combined with LPS stimulation (Figs. 7A and 7B). The same results could be found in lactate levels (Figs. 7C and 7D). Similarly, we examined OCR levels in H9C2 cells by Seahorse assay (Fig. 7E). The results showed that nucleolin interference reduced the OCR levels of H9C2 cells with or without TNF-α combined with LPS stimulation. In our multiple experiments, the data showed that nucleolin interference partially aggravated the decrease in basal respiratory level, reserve respiratory capacity, maximal respiratory capacity, and ATP production capacity after stimulation (Figs. 7F–7I). According to the metabolomic data, it possibly suggested the trend that down-regulated nucleolin could aggravate the decrease in the overall oxidative phosphorylation level of the cells after TNF-α combined with LPS stimulation. Although this trend has been presented many times, there was no significant difference in the effect of nucleolin deficiency on glucose metabolism in response to septic signal stimulation in H9C2 cells. This is a limitation of our experiment. In the future, we will consider expanding the sample size, raising primary cardiomyocytes from mice, or carrying out *in vivo* experiments to further improve our experiment.

Overall, these results further validated the data from metabolomics, demonstrating that down-regulation of nucleolin resulted in glucose metabolism disorder during endotoxemia-induced myocardial injury, while over-expression of nucleolin had the potential to exert a corresponding protective effect.

## DISCUSSION

The heart requires large amounts of energy to maintain its normal function, and therefore the heart needs a continuous supply of energy substrates (*Brown et al., 2017*). Mitochondria are the main consumers of oxygen and the source of ATP, which is produced by the catabolism of a variety of substrates, including fatty acids, glucose, ketone

bodies, and amino acids (*Wende et al., 2017*). However, it is suggested that enhanced glycolytic metabolism contributed to cardiac dysfunction in polymicrobial sepsis (*Zheng et al., 2017*). In the data of metabolomics, we found an elevated rate of glycolysis and significant alterations in the tricarboxylic acid cycle in endotoxemia-induced myocardial injury, which was associated with nucleolin knockout. Our previous experiments revealed that nucleolin myocardial-specific knockout could lead to a decrease in ATP content and aggravated myocardial injury. Therefore, we hypothesized that nucleolin might have affected myocardial glucose metabolism. Therefore, in the present study, we examined the glycolytic function and oxidative phosphorylation in an endotoxemic cell model and demonstrated that down-regulation of nucleolin exacerbated the glucose metabolism disorders caused by endotoxemia-induced myocardial injury. Researchers have already found increased glycolytic pathways and decreased glucose oxidation in sepsis through KEGG annotations and enrichment analysis of differentially expressed metabolites (*Lado-Abeal et al., 2018*; *Pan et al., 2022*), which was consistent with our study. However, our study innovatively suggested that nucleolin might have the ability to regulate this kind of metabolic reprogramming. However, the specific mechanism of nucleolin regulation of glucose metabolism still needs to be further explored.

One of the main steps affecting glucose utilization in cardiomyocytes is glucose uptake, and the relevant molecules are glucose transporter proteins (GLUT1 and GLUT4) (*Standage et al., 2017*). It is also suggested that in septic mice with myocardial injury, increased expression of PDK2 and PDK4 proteins elevated the level of pyruvate dehydrogenase complex phosphorylation, which in turn inhibited its activity, thereby preventing pyruvate from entering the TCA cycle (*Nabben, Luiken & Glatz, 2018*). Therefore, our next experiments should detect whether these molecules related to glucose metabolism have been changed accordingly *in vitro* and *in vivo*, to search for the specific mechanism by which nucleolin regulated glucose metabolism. Additionally, metabolism was also regulated by transcription factors associated with energy metabolism and transporter uptake (*Wende et al., 2017*). The transcription factor PGC-1α not only regulated PPAR-α but also interacted with estrogen-like receptor α (ERRα) and participated in the regulation of glucose and fatty acid transport and ATP synthesis (*Russell et al., 2004*). Our preliminary experiments verified that nucleolin could regulate the transcription factor PGC-1α (*Yin et al., 2023a*, *2023b*), so whether nucleolin affected glucose metabolism through PGC-1α during sepsis-induced myocardial injury still deserves to be explored further. Furthermore, during cellular metabolism, intermediate metabolites produced can participate in the regulation of intracellular signaling and induce cellular translational expression and modification (*Bertero & Maack, 2018*). For example, elevated levels of adenosine monophosphate (AMP) can activate 5′AMP-activated protein kinase (AMPK), and then AMPK activation can promote GLUT 4 translocation (*Li et al., 2019*; *Stanley, Recchia & Lopaschuk, 2005*; *Zaha & Young, 2012*). Interestingly, AMPK has been shown to promote nucleolin phosphorylation (*Gongol et al., 2019*). It is worth exploring whether AMPK, which promotes nucleolin phosphorylation, also plays a role in myocardial glucose metabolism during sepsis.

Enhanced glycolytic metabolism in sepsis might amplify innate immune and inflammatory responses early in the disease and promote immunosuppression in the late stages. It's reported that the enhanced glycolytic metabolism significantly elevated the mortality rate during sepsis, and the use of the glycolytic inhibitor 2-DG to modulate glycolytic metabolism significantly improved cardiac function and survival in septic mice through the regulation of inflammatory response and apoptotic signaling (*Zheng et al., 2017*). Similarly, it has been proved that nucleolin overexpression also had the effect of improving cardiac function and reducing mortality in septic mice (*Jiang et al., 2019*). However, it needed to be verified by more experiments whether nucleolin regulated glucose metabolism and played a protective role through modulating certain immune pathways.

Clinical studies have shown that serum lactate levels lead to a worse prognosis in patients with sepsis, indicating that sepsis and infectious shock were associated with hyperlactatemia (*Chertoff et al., 2015*). Moreover, in the early stages of sepsis, lower serum lactate levels were positively associated with better survival outcomes (*Bakker, Nijsten & Jansen, 2013*). Our findings also further validated that glycolysis was increased and lactate levels were increased in endotoxemia, which could be partially alleviated by nucleolin overexpression but exacerbated by interfering with nucleolin expression. Therefore, we hypothesized that the elevated expression of nucleolin could regulate glucose metabolism to exert cardioprotective effects, which was expected to provide a new therapeutic target for patients with sepsis in clinical practice. However, we recognized the limitations of simulating the cardiac environment in cells when animals are exposed to endotoxemia. To address this, our next step is to establish endotoxemia models in both nucleolin knockout and overexpressing mice to verify our metabolomics findings. Moreover, in the functional experiments, we standardized the experimental data by protein concentration, although we followed the reagent manufacturer's instructions as much as possible, and washed and centrifuged the cells several times during the measurement to remove the interference of the impurities to ensure the accuracy of the experiment. However, the stimulation of sepsis can lead to cell death, resulting in some interference of proteins from dead cells. The interference may be minor, but this is also a shortcoming of our experimental design. In future experiments, we will consider using cell counting to standardize the experimental results, which will be more reliable.

Overall, our data indicated that during endotoxemia-induced myocardial injury, down-regulation of nucleolin promoted myocardial glycolysis and inhibited myocardial oxidative phosphorylation, leading to dysregulation of myocardial glucose metabolism. Given this important role of nucleolin, it is worthwhile to continue the research on the related mechanism, as it would hopefully provide new therapeutic ideas to improve the condition of patients with septic cardiomyopathy.

## ACKNOWLEDGEMENTS

We would like to thank the anonymous reviewers who have helped to improve the article.

### Funding
This work was supported by the National Natural Science Foundation of China (Grant number 81971820). The funders had no role in study design, data collection and analysis, decision to publish, or preparation of the manuscript.

### Grant Disclosures
The following grant information was disclosed by the authors:
National Natural Science Foundation of China: 81971820.

### Competing Interests
The authors declare that they have no competing interests.

### Author Contributions
- Yuting Tang conceived and designed the experiments, prepared figures and/or tables, and approved the final draft.
- Leijing Yin conceived and designed the experiments, prepared figures and/or tables, authored or reviewed drafts of the article, and approved the final draft.
- Ludong Yuan performed the experiments, analyzed the data, prepared figures and/or tables, and approved the final draft.
- Xiaofang Lin performed the experiments, analyzed the data, authored or reviewed drafts of the article, and approved the final draft.
- Bimei Jiang performed the experiments, authored or reviewed drafts of the article, and approved the final draft.

### Animal Ethics
The following information was supplied relating to ethical approvals (*i.e.*, approving body and any reference numbers):

The study protocol was approved by the Department of Animal Experimentation, Medical Ethics Committee of the Third Xiangya Hospital, Central South University (2019-S218).

### Data Availability
The data reported in this article are available at OMIX, China National Center for Bioinformation/Beijing Institute of Genomics, Chinese Academy of Sciences: OMIX005037.

https://download.cncb.ac.cn/OMIX/OMIX005037/

### Supplemental Information
Supplemental information for this article can be found online at http://dx.doi.org/10.7717/peerj.17414#supplemental-information.

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
