# Peer review of "Nucleolin myocardial-specific knockout exacerbates glucose metabolism disorder in endotoxemia-induced myocardial injury"

_PeerJ, doi:10.7717/peerj.17414_

## Round 0.1 · original submission · Major Revisions

Please address the concerns of both reviewers and amend the manuscript accordingly.

**Language Note:** The review process has identified that the English language must be improved. PeerJ can provide language editing services - please contact us at [email protected] for pricing (be sure to provide your manuscript number and title). Alternatively, you should make your own arrangements to improve the language quality and provide details in your response letter. – PeerJ Staff

·

Basic reporting

Although the authors have clearly submitted a large number of findings in a very interesting mouse model in figures 1 and 2, the fonts used are very difficult to read and therefore these figures are hard to assess. In addition, the English should be reviewed for grammar and syntax errors.

Experimental design

How were ATP and lactate measurements normalized? Given that sepsis results in cell death, results should be normalized to living cell counts. In addition, please provide more data for the seahorse methods. What reagents and concentrations are used for each stage, how are results normalized, and how are basal, spare capacity and atp production determined? The O2 consumption methods description appears to describe the use of animal subjects, but only data from H9c2 cells is shown.

Validity of the findings

In figure 4, it is unclear whether the two treatment groups (LPS and TNFa) produce a different effect in the overexpressor or silenced cell groups compared to control. Since the authors’ thesis is that nucleolin impacts the effect of this treatment on ATP production, lactate levels and respiration, this comparison is key. For the seahorse data, by eye it appears that this is the case, although it it not indicated by statistical notation. Please complete these comparisons and add them to the figures. In addition, it is difficult to assess the findings of figures 1 and 2 due to legibility.

Reviewer 2 ·

Basic reporting

The use of the English language in this manuscript is professional and clear. Provided literature references are generally sufficient, and the figures are well-labeled and described. However, I’d recommend improving the quality and resolution of some figures to enhance readability.

Experimental design

The materials and methods section is adequately described.

Validity of the findings

The manuscript demonstrates originality, accompanied by a well-defined problem statement and a clear focus on addressing a gap within the field of study. Although the presented data is sufficiently robust, minor edits in several places are advisable for further enhancement.

Additional comments

I provided some corrections and a few queries to enhance the quality of the manuscript.

Annotated reviews are not available for download in order to protect the identity of reviewers who chose to remain anonymous.

---

## Round 0.2 · Major Revisions

Please address the remaining concerns of reviewer #1 and amend the manuscript in line with the proposed changes.

·

Basic reporting

Figures 1 and 2 are a bit better, but still difficult to read. Font size of all writing (legends, axes, labels, etc.) should be increased so that it is legible when printed out.

Experimental design

Methods detail is significantly improved. However, the authors do not address the issue that sepsis drives cell death and therefore protein normalization may not be sufficient in lieu of a live cell count. An increase in dead cells, which still contain protein, may artificially diminish respiration measurements. Authors should normalize to CCK-8, crystal violet, or trypan blue exclusion counts, or at the very least note this as a weakness in the interpretation of the data. Also, please define the seahorse data calculations within the body of the methods (e.g. state directly that ATP production was determined as the difference in O2 consumption before and after oligomycin, etc.)

Validity of the findings

Given that the authors state in their response that they do not observe a significant difference in ATP, lactate,or any OCR values between control and nucleolin-silenced cells in the presence of LPS and TNFa i, the statements on lines 270-273 “and also aggravated the decrease in basal respiratory level, reserve respiratory capacity, maximal respiratory capacity, and ATP production capacity after stimulation(Figure 4F-I), suggesting that down-regulated nucleolin aggravated the decrease in the overall oxidative phosphorylation level of the cells after TNF-α combined with LPS stimulation” is not supported by the data at this time. The fact that no significant difference between these groups must be clearly stated in the manuscript and discussed. Further, the section 3.4 heading (lines 256-257) is misleading, since metabolism is not significantly altered by nucleolin deficiency during combined stimulation, only in the non-stimulated control groups. Please correct these statements, or repeat the experiments with a sufficient sample size to support the supposition.

Reviewer 2 ·

Basic reporting

No comment

Experimental design

No comment

Validity of the findings

No comment

Additional comments

All my queries were answered satisfactorily. I would recommend the manuscript for acceptance.

---

## Round 0.3 · accepted · Accept

All remaining concerns of the reviewer were adequately addressed, and revised manuscript is acceptable now.

·

Basic reporting

My concerns have been adequately addressed

Experimental design

My concerns have been adequately addressed

Validity of the findings

My concerns have been adequately addressed